# Aerodynamic simulation of rough and eroded blades, AEP effect and mitigation using low drag vortex generators

David Bretos-Arguiñena<sup>1</sup> and Beatriz Méndez-López<sup>1</sup>

<sup>1</sup>Wind Energy Department, Centro Nacional de Energías Renovables (CENER), Sarriguren, Spain **Correspondence:** David Bretos-Arguiñena(dbretos@cener.com)

**Abstract.** Blade roughness depositions or blade erosion have an unquestionably effect over blade aerodynamics and wind turbine power production. This work is focused on the simulation using Computational Fluid Dynamics (CFD) of the NACA 63<sub>3</sub>418 airfoil with high roughness values and with different erosion typologies (pits and extreme losses of material). The CFD code used is OpenFOAM v8 and different technologies are selected to create the meshes to capture properly geometries' defeate (ICEMCED and ANSXS Workbargh)

5 defects (ICEMCFD and ANSYS Workbench).

This study goes a step further by using low drag Vortex Generators (VGs) to mitigate the roughness and erosion harmful effects. Low drag VGs are compared with conventional ones and afterwards 3D blade sections are computed with roughness and erosion incorporating low drag VGs to evaluate the blade performance recovery achieved by the use of VGs. Finally, the impact of the different configurations (rough , eroded , rough + VG and eroded + VG) over Annual Energy Production (AEP)

10 is evaluated on a virtual 2.5 MW wind turbine. The most important finding of the work presented in this paper is that AEP losses due to the existence of blade surface roughness or erosion can be recovered from the use of VGs up to 1.5 %.

# 1 Introduction

Wind turbine blade surface can be degraded during operation by different levels of roughness produced by air particles contamination or initial phases of blade erosion. Even though the effect of roughness over the airfoil performance has worried researchers and industry during the last 20 years, the 2D quantification of the effect that high roughness levels has over airfoil performance is still unresolved. With regard to erosion, there is not a standardized methodology to estimate its effect over airfoil performance or wind turbine AEP.

Roughness distribution on airfoil surface affects its performance by degrading the airfoil aerodynamic behaviour. If rough elements are located near the leading edge zone they influence the laminar to turbulent transition process moving upwards the transition point. On the other hand, roughness also modifies the flow characteristics in fully turbulent flows. Both effects are often confused with each other and the second is sometimes ignored. The deposition of big roughness elements on the blade surface is considered equivalent to the first stages of blade erosion. In addition, the growing installation of offshore wind farms

and the aim of reducing the cost of energy lead to the necessity of predicting accurately the energy losses due to rough or eroded blades in order to decide when to undertake wind turbine erosion control strategies or blade reparations.

The accurate prediction of the effect of surface roughness over the airfoil aerodynamic behaviour is of great interest for engineers, specially for the design of wind turbine blades. Aerodynamic predictions of airfoils sensitivity to roughness is an extremely challenging issue. Standish et al. (2012) performed experiments and simulations of a non specified airfoil of 18% thickness. The simulations were made using ANSYS CFX and roughness effects were accounted for by introducing an

- equivalent sand-grain roughness on surface patches. Afterwards AEP was calculated to evaluate the effect of roughness on power production. Normalized mechanical power output relative to measured power at two operational points below rated power of the SWT-2.3-93 wind turbine lead to power output reduction in 6 % and 7 %. Ehrmann et al. (2013) presented a study on the effect of leading edge roughness effect on airfoil performance. Different roughness types where characterized through measurements with laser scans on in-service wind turbine blades to perform wind tunnel simulations of a NACA
- $63_3418$  airfoil. The study was completed with computations using the flow solver OVERFLOW with a roughness amplification model for the Langtry-Menter transition model implemented. Differences between drag were observable between the clean and paint roughness configurations. More recently, further research has been done on the the field. Bak et al. (2020) studied the influence of the wind characteristics on the power production loss due to the existence of imperfections on the airfoil surface. It was concluded that the relative AEP losses due to leading edge damages reduced with increasing average wind speed on sites.
- For the low wind speed site the losses were between 1 % and 4 % depending on the extent and the type of the blade damage. For the high wind speed site the losses were between 0.5 % and 3 %. Furthermore, the bigger the extent of the damages the bigger the losses were. In addition, it was shown that increasing the maximum tip speed increased AEP and decreased the losses. Maniaci et al. (2020) focused their studies on reducing the uncertainties and AEP losses of 3% were reported when erosion is present in the National Rotor Testbed. Kruse et al. (2020) published CFD simulations of the NACA 63<sub>3</sub>418 and
- compared with experiments both with sandpaper and zz tape. Roughness was simulated with the Knopp model and the main conclusion was that not the lift decrease neither the drag increase were well captured. More recently, the authors of this paper presented detailed simulations of the NACA 63<sub>3</sub>418 using OpenFOAM Méndez and Pires (2022) for several roughness values and compared them with the wind tunnel experiments performed in the Second Call of Joint Experiments project described in Pires (2018).
- Moving on to the study of blade erosion effect over airfoil performance simulations some work has been performed in recent years. Kruse et al. (2021) evaluated the NACA  $63_3418$  airfoil with 1000 different protuberances in the airfoil leading edge ,the main conclusion was that position and the depth/height of the disturbance, with up to 35% lift reduction and 90% lift/drag reduction within the specified angle of attack (AoA) and disturbance parameter ranges. Campobasso et al. (2022) presented a study in which large and sparse erosion cavities where simulated in a DU96W180 airfoil for a Reynolds number of  $1.5 \cdot 10^6$ .
- It was found that the considered cavities can trigger transition, indicating the necessity of both resolving their geometry in the simulations and also modeling distributed surface roughness, since it affects the boundary layer characteristics and may trigger transition over the entire spanwise length affected. The energy yield loss of NREL 5MW wind turbine due to the considered erosion pattern is found to be between 2.1 % and 2.6 % using measured and computed force data for the nominal and eroded

outboard blade airfoil. A parametric analysis of the cavity geometry suggests that the geometry of the cavity edge has a much
larger impact on aerodynamic performance than the cavity depth. Saenz et al. (2022) measured eroded blades after several years of operation and performed a statistical study of the different sizes, shapes and locations of erosion. They were studied using CFD and the effect over the NREL 5 MW computed. The conclusion was that as closer to the leading edges and as sharper the erosion corners the most harmful effect over AEP was obtained and quatified as 3% reduction. Vimalakanthan et al. (2022) presented combined transition and rough CFD simulations of NACA 63<sub>3</sub>418 airfoil and the CFD modelling of an actual eroded blade. The conclusion of the study was that the calibrated CFD model for modeling flow transition accounting roughness shows good agreement of the aerodynamic forces for airfoils with leading-edge roughness heights in the order of 140-200 µm when comparing with the experiments, while showing poor agreement for smaller roughness heights in the order

of 100  $\mu$ m. The study indicated that up to 3.3 % reduction in AEP can be expected when the LE shape is degraded by 0.8 % of the chord, based on the NREL 5 MW wind turbine.

- Besides the studies needed to characterize the effect of roughness or erosion over airfoil performance, mitigation measures with aerodynamic devices have been designed and optimized to improve blade performance. One of the most used aerodynamic devices for this purpose are Vortex Generators (VGs). VGs were used traditionally in the root area of the blades to improve the performance of the thick airfoils located in that zone. More recently vortex generators are being included in the tip area of the blades to improve efficiency in rough or eroded conditions. VGs are designed to create vorticity on the blade surface
- mixing high momentum zones of the upper part of the boundary layer with low momentum zones near the surface resulting in a velocity profile less prone to separation. Gutiérrez et al. (2020) worked on the definition of mitigation measures to reduce the harmful effect of roughness and an experimental study was presented. It was concluded that the increment in drag produced by VGs is negligible compared to the already present by roughness. There is not a significant change in drag due to the usage of VGs on the pressure side Hansen et al. (2016) proposed an aerodynamically shaped VG that was manufactured and tested in
- the wind tunnel. It was added to a DU-91-W2-250 airfoil and an efficiency increase was observed with regard to conventional vortex generators from 3.6 % to 16.36 %. With regard to simulation methodologies, when simulating VGs both fully resolved and modelling approaches are used. Seel et al. (2022) presented a comparison of performance of the use of Bay model versus the fully resolved approach when computing VGs. The agreement in terms of lift and pressure distribution is very good whereas the drag is underestimated by the BAY model
- In terms of power production, Fernandez-Gamiz et al. (2017) studied in their work the energy production increase of NREL 5 MW wind turbine using the blade element momentum theory and concluding that an overall increase of 3.85 % on the average wind turbine power output can be found when using VGs and gurney flaps. These VGs where located in the blade root and mid span area and no erosion or surface roughness were included in the simulations.

Skrzypiński et al. (2020) studied the effect over AEP of retrofitting blades affected by surface roughness using VGs and 90 determined experimentally a gain of 3.3 % and 2.8 % predicted with and engineering tool.

This study is organized as follows: first, CFD simulations of the NACA  $63_3418$  airfoil have been performed in several surface conditions (clean, rough, pits and extreme loss of material). Then, CENER's low drag VGs are described and the CFD simulations of blade sections for the studied surface conditions including VGs are shown. Finally, the annual energy production

of a reference 2.5 MW wind turbine is evaluated to define the effect of roughness or erosion over AEP and to estimate how 95 much of these losses can be recovered by the installation of low drag VGs over the blade surface.

#### 2 CFD simulations of rough and eroded blades

In the following subsections, simulation results for a large roughness level case and different typologies of erosion are exposed. The different simulations results are compared in terms of blade section performance.

#### 2.1 NACA $63_3418$ airfoil CFD simulations for a roughness level of P40 (423 $\mu$ m)

- In this section, the simulations for the NACA  $63_3418$  airfoil are presented for  $Re = 3 \cdot 10^6$ . The simulations were performed in clean and rough blade surface conditions with OpenFOAM v8 and the mesh was created using ICEMCFD. The number of nodes in the airfoil are 434 and the minimum size of the elements close to the surface is  $5e^{-6}$  m for the clean cases mesh in order to obtain y+ value below 1 to ensure the proper performance of the turbulent model. On the other hand, for the rough cases, the y+ values close to the airfoil surface are around 30, these y+ values are needed for the correct performance of the
- roughness model. The roughness model used is the one included in OpenFOAM v8 based on Cebeci (1977). This model applies modified logarithmic wall functions for y+ over 11.25. Going up to a y+ of 30 avoids the uncertainty of being in the buffer layer region which doesn't present a logarithmic behaviour with regards to U+.

Clean cases are computed using the eN and the k-kl-w transition model. The eN transition method from van Ingen (2008) to simulate clean conditions is implemented in OpenFoam v8 by modifying the turbulence model intermittency factor and

- transition location for each angle of attack is imposed trough a connection with the panel method XFOIL Drela (1989). The k-kl-w transition model is the one available in OpenFoam Walters and Cokljat (2008). In the rough cases the airfoil chord covered with rough elements is 15 % both on lower and upper sides on the leading edge zone. The roughness selected is P40 (423  $\mu$ m) using in this case the equivalent sand grain value of  $h_s = 5e^{-3}$  to compare with the experimental results measured in the IRPWind Joint Experiments Project Pires (2018). A more extended evaluation of rough simulations was presented in
- Méndez and Pires (2022) where several roughness models and roughness sizes were evaluated. These polar curves for the clean and rough cases will be used to compute the AEP for a reference wind turbine in Section 4. Figure 1 compares the airfoil lift coefficient, drag coefficient and efficiency both in clean (transitional computation with the k-kl-w model) and rough cases ( comparing computations and experiments). It is observed that the CFD results reproduce accurately the experiments. A light over-prediction of the stall area in the lift coefficient for the clean conditions is observed.