# Peer review of "Aerodynamic simulation of rough and eroded blades, AEP effect and mitigation using low drag vortex generators"

_Wind Energy Science, 2023_

## Referee Comment (RC2)

Review of the manuscript:

*David Bretos-Arguiñena and Beatriz Méndez-López:*

**"Aerodynamic simulation of rough and eroded blades, AEP effect and mitigation using low drag vortex generators"**

**General comments:**

The manuscript presents calculations on the influence of leading edge erosion on the performance of blade segments and discusses the effect of vortex generators (VGs) to mitigate the performance reducing erosion effects. Aerodynamically shaped VGs are presented, for which a higher performance is postulated compared to conventional VGs. The authors do not analyze the reason for this advantage and some questions remain unanswered (see below). For a blade segment with these VGs, polars are calculated by means of RANS simulations, which are then used to determine the influence of erosion and VGs on the AEP for a generic wind turbine.

The topic is relevant and the approach to reduce erosion-induced power losses by VGs is interesting. Unfortunately, the manuscript as presented does not meet the requirements for a scientific publication in WES. The flow physics analyses and explanations of the effects of the suggested VGs are almost completely missing, some results seem unplausible without further explanations and analyses, the manuscript is obviously written quickly and contains many typos, linguistic errors and only few literature sources. Overall, sound analyses have to be supplemented and the manuscript has to be completely revised to be considered for publication in WES.

**Specific comments and remarks:**

- The manuscript contains many unclear and linguistically incorrect formulations and needs to be completely revised.

- Abstract: The abstract is essentially a short summary and only the last sentence mentions a finding from the study. The presentation of the findings should be strengthened. In the abstract the calculation method (steady RANS?) should also be mentioned and it should be explained which "different technologies" are used in the calculations. Furthermore, it should be clarified in which respect the study goes "one step further".

- Line 7: "Conventional VGs" should be further specified (height, shape, …). In the remainder of the manuscript, conventional VGs are not distinguished regarding their shape although they have specific characteristics according to the current state of research.

- Line 16: Some references on VG studies done in the last 20 years should be provided.

- Line 21: In which way does roughness modify the flow characteristics of a fully turbulent flow? I would rather use "impacts" or "affects" instead of "modifies".

- Line 22f: Can the authors provide evidence (in form of references) that roughness elements act like first stages of blade erosion?

- Linie 29: In the manuscript often just the name of the CFD code is mentioned (L.35, L.47). The authors should also mention the method used (steady-state RANS? turbulence model?) to enable the reader to better assess the work. The term "CFD" is too general.

- Line 32: The statement of a power reduction of 6-7% is too general. For which roughness height was this value determined and was a fully turbulent flow or a natural transition scenario considered?

- Line 37: „on the field". Does this refer to "field measurements" or investigations in the area of simulating roughness effects?

- Line 45: Please insert a reference for the "Knopp" model.

- Line 51: What was varied with the 1000 different protuberances? Height, position, spacing, shape? The sentence is grammatically not correct.

- Line 64: The reference is not up to date, compare current status on WES (published 2023).

- Line 64/65: What are "combined transition and rough CFD simulations"? What does "actual eroded blade" mean? Was the actual shape of the leading edge erosion measured and taken into account in the CFD simulation? Which transition model was calibrated?

- Linie 70ff: The wording implies that VGs were developed for erosion problems. This sentence should be revised.

- Line 73/74: Please add a reference for this statement.

- Line 74: I suggest: "streamwise" vorticity

- Line 74: „on the blade surface" → "within the blade boundary layer"

- Line 77/78: „Increment in drag" compared to what? Compared to a natural tranistion scenario?

- Line 78/79: Sentence is unclear. Is meant that drag is not increased when VGs are placed on the lower side (which would surprise me) or is meant that the drag of the lower side is not increased by VGs attached to the upper side? The dot at the end of the sentence is missing.

- Line 79-81: The authors should describe in some more detail how Hansen et al. designed their aerodynamically shaped VGs and why the efficiency increase is spread over such a wide range. As a part of the literature review of the present manuscript it should additionally be mentioned that the $(L/D)_{max}$ of the clean airfoil is significantly higher than that of the airfoil equipped with VGs (even for the aerodynamically shaped ones).

- Line 82: Dot missing at end of sentence. Please use "Bay" or "BAY" consistently. What are the advantages of the fully resolved simulation of VGs and why did the authors choose this approach?

- Linie 82: The co-author published a study on low-drag VGs in 2018. Why was this paper not referenced and used as a basis for the present study? (doi: 10.1088/1742-6596/1037/2/022029)

- Line 86: Please be more precise. The power estimation was performed with BEM using (2D? calculated or measured?) polars for the blade sections without and with VGs/gurneys.

- Line 89//90: Please revise this sentence.

- Line 91ff: Please refer to the section numbers of your manuscript.

- Line 100ff: The description of the numerical model is considerably too scarce to evaluate its suitability for the present analyses. How many grid cells were introduced across the boundary layer height? Number of total grid cells, mesh topology, far field distance, turbulence model, steady RANS or unsteady URANS, numerical scheme...? In the remainder of the manuscript unstructured and structured meshes are mentioned and a comparison is shown in Fig. 2. To the reader it is not clear which calculations documented in the manuscript were performed with which mesh type.

- Line 101ff: "in clean" → "for clean"; "in the airfoil" → "along the airfoil"; throughout the manuscript: "y+" → "$y^+$"; "turbulent model" → "turbulence model"; "ICEMCFD" → "ICEM CFD"

- Line 103: Please first mention that a wall model is used for the rough case, then the requirements for $y^+$.

- Line 108 and throughout the manuscript: „eN" → "$e^N$", "k-kl-w" → "k-kl-$\omega$". Remark: In XFOIL a simplified $e^N$ envelope method is implemented, in which the frequency dependence of the TS amplification is not considered, in contrast to the original method proposed by van Ingen.

- Linie 113: 5e^{-3} → $5 \cdot 10^{-3}$. This should be used consistently throughout the manuscript.

- Line 118: The experiments are well reproduced by the calculation except for the stall domain.

- Caption Fig. 1 and Fig. 2: The CFD results are only shown as lines, so that the chosen angle-of-attack discretization is not clear. I suggest to add some symbols.

- Caption Fig. 2: „eN turbulence models". The $e^N$ method is not a "turbulence model" but rather a "transition prediction model".

- Fig. 3: In the legend (XFOIL) I suggest to write "transition location upper / lower side (XFOIL)."

- Line 130: „shows the pressure distribution" or „shows the distribution of the pressure coefficient"

- Line 149: „SST k-w" → „SST k-$\omega$". Was this turbulence model also used in the previous calculations with prescribed XFOIL predicted transition location? Then this should be mentioned in Sec. 2.1.

- Figs. 10/15/16/18: Are the velocities in the first grid layer shown in the top views? Is the axial velocities component shown or the absolute value of the velocity? The legends in the top views are too small and partly blurry.

- Table 1 / Fig. 8: ($\alpha$ in [º], $\Delta(C_l/C_d)_{max}$?!). Obviously, the calculations were only performed up to an angle of attack of $14^0$ and at least for the clean and the turbulent case without erosion, the theoretical $c_{l, max}$ values are obviously not yet reached. This angle should therefore not be listed as $\alpha(c_{l, max})$ in Table 1.

- Line 167f: VGs only increase $c_l$ in case of flow separation for the baseline airfoil, and particularly increase $c_{l, max}$ and $\alpha(c_{l, max})$. They do not increase lift for the complete AoA range. This should be expressed more clearly.

- Line 168: I suggest: "… added to the blade for attached flow".

- Line 169ff: CENER proposes VGs whose cross-section feature a 10% thick aerodynamically shaped airfoil. Given the very small Reynolds numbers of the VG, flat plates are actually not a bad choice and provide well defined separation locations for the generation of the streamwise vortices. The authors initially claim without evidence that their aerodynamically shaped VG cross section show lower drag than conventional VGs, but it is not clear whether this statement refers to the self drag of the VGs or to the drag of the blade section equipped with these VGs. The authors do not provide an explanation or a reference. Fig. 14 then seems to support the claim. It is surprising, however, that the airfoil with the proposed VGs shows a significantly lower drag than the turbulent airfoil without VGs, even at small angles-of-attack. This is not plausible to me and is also in contrast to previous studies. For example, it was shown by Hansen et al. (doi: 10.1002/we.1842) that for the (aerodynamically shaped) VGs they studied, the $L/D_{max}$ of the blade section with VGs is reduced. In the present manuscript a sound flow-physics analysis and a physical explanation of the beneficial effect of the proposed VGs is urgently needed! Does the turbulent airfoil without VGs show flow separation even at small angles of attack, which are suppressed or delayed by the VGs? If so, how can it be explained that the conventional VGs do not suppress the separation? Are mean momentum and displacement thicknesses of the boundary layer reduced by the VGs? Here, an analysis of the boundary layer development without / with VGs as well as an analysis of the flow field and of skin friction and pressure drag components is necessary.

- Line 174: The aerodynamic behavior of VGs depends primarily on the geometrical parameters of the VG array (i.e., inclination angle, interspacing, intraspacing, VG height with respect to the local GS thickness, VG length on the airfoil surface and at the VG tip, positioning of the VG pairs with respect to each other). This should be mentioned in the paper. If necessary, the influence of these parameters should also be investigated. Hansen et al. showed that the inclination angle still produces a large increase in efficiency. Is this also the case with the CENER VG?

- Fig. 14: In this graph, the difference between conventional (Delta, Cropped and Rectangular) VGs is not visible. Can you zoom in to show the differences? Differences of conventional VGs should be predicted in qualitative agreement to results of previous studies to make the predictions for the proposed VGs more plausible.

- Line 184: Was the boundary layer of the VGs actually resolved in the present calculations? What criteria were used to ensure accurate meshing of the VGs? It is unclear how a meaningful refinement in the VG region was achieved without increasing the total number of grid cells.

- Linie 197f: The parametric study of the chordwise VG position for rough conditions should be shown in this work. Since the boundary layer profiles are significantly changed by roughness, it may not be feasible to use best practices for clean airfoils. A detailed study of the boundary layer parameters would add scientific value.

- Linie 197ff: Regarding their efficiency the different types of conventional VGs are not distinguished in this statement, which does not reflect the current state of research (see Schubauer et al. https://doi.org/10.1017/S0022112060000372 or more recently Baldacchino et al. 10.1002/we.2191).

- Linie 199f: Vortex formation of VGs is a complex topic, which cannot be covered by seemingly arbitrarily placed streamlines. More meaningful properties (velocity profiles in the wake, boundary layer parameters or cross-flow kinetic energy (see dissertation Florentie https://doi.org/10.4233/uuid:704d764a-6803-4cad-991f-45dc4ea38f6d)) should be used here.

- Fig. 15: What is the meaning of this figure, the text does not contain any interpretation? Why is the flow field around the new VGs shown in a different way in Fig. 16? Here, a representation as in Fig. 15 a) b) c) and a direct comparison with the flow fields of the conventional VGs in connection with a flow-physics analysis would make sense. What is the origin of the geometric parameters of the three configurations? Is the thickness of the VGs to scale?

- Fig: 16: The figure does not add value as it is not suitable for assessing the low drag VG. Quantitative analysis should be provided here, e.g. as presented by the co-author at Torque 2018 (doi :10.1088/1742-6596/1037/2/022029).

- Line 208/209: The statement "...recovery of the airfoil efficiency..." is exaggerated. The improvement in performance is actually very small. Furthermore, I cannot understand the values given in Table 2 for $\Delta(C_l/C_d)_{max}$. According to the table, config. 2a should show a L/D reduction of more than 32% compared to turbulent flow, which I cannot see in Fig. 17. I ask the authors to check and correct the values listed in table 2.

- Fig. 18 / 19 and associated text: How were the separation locations (65% and 75%) determined, for which spanwise range are the $c_f$ values shown in Fig. 19 and could you show spanwise averaged distributions? What is the reason for the strongly positive or negative $c_f$ values at 30% chord? How can a single small VG pair have such a strong influence on separation location of a blade section with such a large span? More generally, I also wonder why such a large span was considered in the calculations, but only one VG pair modelled. Would it not have made more sense to use a segment with a smaller width and periodic boundary conditions?

- Line 226: What are the „corresponding polar curves"?

- Sec. 5: Conclusions consisting only of a list are unusual. Some of the aspects mentioned were not discussed in the manuscript and may not be included in the conclusions, for example line 247-249. Other listed conclusions actually represent an outlook: line 255-256, line 262-264. The conclusion line 261-262 was not discussed and verified in the manuscript.

**Typos:**

- Line 33 and throughout the manuscript: „where" → „were"

- Line 64: "quatified" → "quantified"

- Line 126: "tree" → "three"

- Line 145: "show" –"shows", "in the surface" → "on the surface"

**Linguistic inaccuracies:**

- Line 15 and throughout the manuscript: "over the performance" → "on the performance"

- Line 97 and throughout the manuscript: "typologies" → "types"

- Line 132: "the pressure over the airfoil surface value". Please revise.

- Line 136/137: please revise "inspired in", "consist in pit and gauges".

- And many more

---

## Author Comment (AC1)

**Reply on CC1**

David Bretos-Arguiñena[1] and Beatriz Méndez-López[1]

[1]Wind Energy Department, Centro Nacional de Energías Renovables (CENER), Sarriguren, Spain

**Correspondence:** David Bretos-Arguiñena(dbretos@cener.com)
* * *
We thank the reviewer for the time invested reviewing the manuscript and for their comments to further enhance the content of the manuscript. The different suggestions are explained below and the manuscript has been extended to create the second version.

1. **Could you please provide a glossary of terms, symbols and include definitions for acronyms or abbreviations used throughout the paper**

   In the manuscript, the first time that one acronym was used in the text, it was given it's definition. But in order to make it's search easier and to answer to the requested suggestion, a glossary of terms and symbols including it's definitions is provided at appendix A. Thanks to the reviewer for this suggestion.

   **Appendix A:  Abbreviations**

| | |
|---|---|
| AEP | Annual energy prediction |
| AoA | Angle of attack |
| BEM | Blade Element Moment |
| c | Chord length |
| CFD | Computational fluid dynamics |
| Cd | Drag coefficient |
| Cf | Skin friction coefficient |
| Cl | Lift coefficient |
| Cp | Pressure coefficient |
| $h_s$ | Equivalent sand-grain roughness |
| $im$ | Intermittency |
| $k$ | Turbulence kinetic energy |
| $kl$ | Laminar turbulence kinetic energy |
| $kt$ | Turbulent turbulence kinetic energy |
| $nut$ | Turbulent viscosity |
| RANS | Reynolds-Averaged Navier Stokes |
| SIMPLE | Semi-Implicit Method for Pressure-Linked Equations |
| SST | Shear stress transport |
| S-M | Structured mesh |
| Re | Reynolds number |
| $p$ | Pressure |
| $U$ | Velocity |
| U-M | Unstructured mesh |
| $U^+$ | Dimensionless velocity |
| $\omega$ | Turbulence specific dissipation rate |
| $y^+$ | Dimensionless wall distance |
| VG | Vortex generator |

2. **On page 3, line 73 –Any reference to support your statement: "More recently vortex generators are being included in the tip area of the blades to improve efficiency in rough or eroded conditions."**

Thanks for your comment. This sentence is supported by the chart on slide 3 of the presentation made by Jesper Madsen (LM Wind Power) in the Annual Event 2017 of Wind Energy Denmark. In that slide, it is stated that LM´s vortex generators could be installed in the first 90% of the blade span, and in the tip area to mitigate aerodynamic impact of eroded blades.

This reference has been added to the manuscript´s references list.

3. **On page 4, line 102 –Please include the chord length of the airfoil**

Thanks for the suggestion, it has been included in the manuscript in the following way:

> The blade chord is 1 m, the number of nodes in the airfoil are 434 and the minimum size of the elements close to the surface is $5e^{-6}$ m for the clean cases mesh in order to obtain y+ value below 1 to ensure the proper performance of the turbulence model.

4. **On page 4, line 108 – What value of Ncrit was used for the eN transition method? Please include in text**

A Ncrit=9 value was set, it has been included in the manuscript in the following way:

> The eN transition method from van Ingen (2008) to simulate clean conditions is implemented in OpenFoam v8 by modifying the turbulence model intermittency factor and transition location for each angle of attack is imposed trough a connection with the panel method XFOIL Drela (1989). In order to get the upper and lower transition locations with XFOIL a $N_{crit}$=9 was set.

5. **On page 4, line 109 – The CFD model setup description is insufficient, could you please provide more information on convergence criteria, boundary conditions, force and residual history plot? and the order of models and solvers used for the computation. Please specify the residual drop, i.e. by how many orders of magnitude the residuals dropped for all equations. This is important if the order of magnitude of initial residuals is not 1 for all PDEs (i.e. are these residuals normalized?).**

We completely agree with the reviewer and we find necessary to add that information in a new section as follows:

The OpenFOAM CFD code was employed for the presented simulations. Steady-state time scheme and RANS (Reynolds-Averaged Navier Stokes simulations) turbulence models were used for all the computations performed. The incompressible solver *simpleFoam* based on SIMPLE algorithm (Semi-Implicit Method for Pressure-Linked Equations) of Caretto et al. (1973) was used.

The selected solvers and smoothers for solving the different terms equations are shown in the Table 1.

**Table 1.** Selected solvers and smoothers.

| Equation | Solver | Smoother |
|---|---|---|
| $p$ | Geometric-algebraic multi-grid (GAMG) | Diagonal incomplete-Cholesky (DIC) |
| $U$ | | |
| $k$ | | |
| $\omega$ | Solver using a smoother (smoothSolver) | Gauss-Seidel (GaussSeidel) |
| $kl$ | | |
| $kt$ | | |

In this article different turbulence models have been used in order to consider different flow conditions, these turbulence models are introduced in the following sections. However, the assigned boundary conditions associated with the different variables employed by each model are shown in Table 2 and their meaning is explained at Appendix A.

**Table 2.** Assigned boundary conditions for the different performed simulations.

| Patch | $U$ | $p$ | $k$ | $\omega$ | $nut$ | $kl$ | $kt$ | $im$ |
|---|---|---|---|---|---|---|---|---|
| Farfield | freestream | freestream | inletOutlet | inletOutlet | calculated | fixedValue | fixedValue | inletOutlet |
| smoothAirfoil | fixedValue | zeroGradient | wallFunction | wallFunction | wallFunction | fixedValue | fixedValue | fixedValue |
| roughAirfoil | fixedValue | zeroGradient | wallFunction | wallFunction | roughWallFunction | fixedValue | fixedValue | fixedValue |

In the different simulations a fixed number of iterations was established, in order to achieve convergence for high angles of attack, since these are the most problematic ones. For the convergence criteria, the stabilisation of aerodynamic coefficients ($C_l$, $C_d$ and $C_m$) was used. The main reason to choose this criterion is to avoid the early ending of the simulations while the aerodynamic coefficients keep still changing due to a bad selection of residuals' values, and also to avoid misleading cases of false convergence due to excessively low relaxation factors. Once the different coefficients values keep oscillating around the same value, the simulations were considered converged.

Although residuals were not considered as convergence criteria, they were reviewed at some simulations (Fig. 1) in order to do not omit any strange behaviour. The obtained residuals are normalized ($L_1$ norm) and scaled (the initial residual values is 1). Residuals drop varies depending on the angle of attack, the larger the angle of attack is, the lower the residuals drop is. Regardless of the size of residuals' drop, the aerodynamic coefficients converge. For high angles of attack the aerodynamic coefficients oscillate remarkably due to the non stationary flow nature.

[Figure]

**Figure 1.** Residuals and aerodynamic coefficients evolution during the simulations for angles of attack of 0º, 12º and 20º; and the SST k-w turbulence model.

6. **On page 4, line 113 – What relationship was used to determine the equivalent sand grain value, please provide a reference**

   Thanks for the comment, the following information and references have been added to the manuscript.

   The equivalent sand grain approach links the real roughness $h$ to an idealized roughness with reference to Nikuradse's experiments . The height of the equivalent sand-grain, $h_s$, is deduced from the real roughness shape with the help of the empirical correlations proposed by Dirling and Grabow and White . These correlations are summarized next:

$$\frac{h}{h_s} = \begin{cases} 60.95\Lambda^{-3.98} & \text{for } \Lambda < 4.92 \\ 0.00719\Lambda^{1.9} & \text{for } \Lambda > 4.92 \end{cases} \tag{1}$$

with $\Lambda$ being the roughness shape parameter defined as

$$\Lambda = \frac{l}{h}\left(\frac{A_s}{A_p}\right)^{4/3} \tag{2}$$

with $l$ being the average distance between the roughness elements, $h$ their average height, $A_p$ the element frontal area and $A_s$ the element wet surface. The $h_s$ parameter is dependent of the real roughness elements size and the area covered and would be the one used in this study to evaluate the roughness effect on the airfoil aerodynamics.

Nikuradse J. Strömungsgesetze in rauhen Rohren (Laws of flow in rough pipes - NACA TM 1292) VDI-Forschungsheft, Tech. Rept. 361, 1933

Dirling Jr R.B. A method for computing rough wall heat transfer rates on reentry nosetips Proceedings of the 8th Thermophysics Conference, AIAA Paper 73-763, 1973.

Grabow R.M. and White C.O. Surface roughness effects on nosetip ablation characteristics AIAA Journal, Vol 13, 1975

7. **Page 5, line 121 – Please provides some images of the difference between the (S-M) and (U-M) grids**

Thanks for the comment. A new figure showing the structured and unstructured meshes is included.

> In order to justify the selection of the k-kl-w transition model as baseline in this work a comparison of the eN and k-kl-w method is presented in Fig. 3 comparing with the experiments available from Pires (2018). In addition structured meshes (S-M) and unstructured meshes (U-M) are compared, Fig. 2 shows the difference between both types of meshes.

[Figure]

**Figure 2.** Meshes for NACA $63_3418$: (a.1) S-M main view, (a.2) S-M leading edge, (a.3) S-M trailing edge, (b.1) U-M main view, (b.2) U-M leading edge, (b.3) U-M trailing edge.

.

8. **Page 15, Figure 13 – Please include the chord location of the VGs in the caption**

Thanks for the suggestion, it has been included in the manuscript in the following way:

[Figure]

**Figure 13.** Mesh for rough (left), type 1 eroded (middle) and type 2 eroded (right) cases with CENER´s Low Drag VGs (30 %c).

9. **Page 15, Line 194 - Please provide a table with the dimensional information of the different VGs, such as the vane angle, the lateral distance between the vanes and height with respect to the chord and/or local boundary layer thickness**

   Thanks for the suggestion. The dimensional information of the VGs is included in the new version of the manuscript. The VG height is 5 mm, their chord is 12.5 mm and the angle of attack to the incoming flow is 18 degrees. The separation between the VGs trailing edge is 12.5mm.

10. **Page 15, Line 195 – What was the metric used to determine the best VG location along the airfoil chord, please provide the results from the parametric study.**

    Thanks for the comment, we agree that it is very interesting to add this information to the manuscript.

    The code presented by Delphine du Tavernier in her work has been used to make a parametric study that helps to select the best location of the VG pair. This reference has been added to the manuscript and the description of how the best location for the VGs has been included. The best location has been selected for the clean airfoil case, so it is assumed by the authors that for the other configurations the 30% value may change. It has been maintained throughout the work in order to compare the AEP values obtained in the last part of the paper.

    Tavernier, Baldachino, Simao 'An integral boundary layer engineering model for vortex generators implemented in XFOIL' - Wind Energy (https://doi.org/10.1002/we.2204), 2018

11. **Page 21, Line 225 – Please provide more detail about the AEP model, was it via BEM calculations? was drag contribution added to the induction calculation? Which wind distribution was considered, and which mean wind-speed was used?**

    Thanks for the suggestion, we appreciate it and the section will be expanded in the manuscript adding a more detailed description of the code BladeOASIS and including a reference to a publication in which the code has been described. In addition the wind condition will explained: stationary wind is used and the Weibull parameters are c=7.5 m/s and k=2. In addition, the process used to include the CFD results has been elaborated in detail (mainly the computed polar curves are used in the BEM part of BladeOASIS). The tip loss correction from Glauert has been used.

---

## Author Comment (AC2)

**Reply on RC1**

David Bretos-Arguiñena[1] and Beatriz Méndez-López[1]

[1]Wind Energy Department, Centro Nacional de Energías Renovables (CENER), Sarriguren, Spain

**Correspondence:** David Bretos-Arguiñena(dbretos@cener.com)

We thank the reviewer for the time invested analyzing the manuscript and for their comments to further enhance the content of the manuscript. The paper has been changed attending the referee suggestions, the typo errors corrected and more descriptions added. In addition, the answers to the different questions are summarized next.

1. **The novelty of the paper should be listed.**

   Thanks for the suggestion. The novelty of this work is based on three main pillars:

   – Advance in meshing techniques to introduce erosion and vortex generators simulating accurately the boundary layer without high loaded meshes

   – Demonstration of the advantages of low drag vortex generators versus conventional ones

   – Assessment of the capacity of vortex generators of annual energy recovery in rough or eroded blades

   This facts have been addressed in detail in the new version of the manuscript.

2. **You need to cite references for your statements in the introduction. Your discussion in the introduction goes too fast to 2D quantification. A general discussion about blade roughness and erosion for wind turbine blades is interesting.**

   Thanks for the comment. The introduction has been expanded in the manuscript to improve the initial discussion quality. In addition, new references have been added in many of the topics.

3. **"moving upwards": what direction is upwards?**

   We agree with the reviewer that the selected expression for this statement is not clear enough, this expression has been replaced by the following one in the manuscript:

   > Roughness distribution on airfoil surface affects its performance by degrading the airfoil aerodynamic behaviour. If rough elements are located near the leading edge zone, laminar to turbulent transition upper and lower points are displaced towards the leading edge.

4. **Line 28: What is "non specified airfoil"?**

20    The only geometrical data specified for the airfoil at Standish et al. (2012) is the relative thickness (18% airfoil thickness). The airfoil's nomenclature or airfoil family are not mentioned. However, we also consider that the expression "non specified" is confusing and we have changed it by "unknown" on the following way:

> Standish et al. (2012) performed experiments and simulations of an unknown airfoil of 18% thickness. The simulations were made using ANSYS CFX and roughness effects were accounted for by introducing an equivalent sand-grain roughness on surface patches. Afterwards AEP was calculated to evaluate the effect of roughness on power production.

5. **Line 103: What turbulence model is "the turbulence model"?**

Thanks for your suggestion, we agree that this sentence is not clear. It has been changed in the manuscript by the
25    following one:

> The blade chord is 1 m, the number of nodes in the airfoil are 434 and the minimum size of the elements close to the surface is $5e^{-6}$ m for the clean cases mesh in order to obtain y+ value below 1 to ensure the proper performance of the turbulent (SST k-w) and transitional (k-kl-w and eN) models.

6. **Section 2.1: What is k-k1-w transition model? A short description with reference is required. Same comment for the transition model of eN.**

Thanks for your comment, a more elaborated description of the models used are included in the manuscript. The k-kl-w transition model (Walters and Cokjat 2008) is an eddy-viscosity turbulence model of three transport equations (the
30    specific dissipation rate $w$, and the turbulent $k_T$ and laminar $k_L$ kinetic energy equations) based on k-w turbulence model.

The eN transition model is a semi-empirical method based on linear stability theory suitable for transonic and low Reynolds number airfoils. The upper and lower transition points from laminar to turbulent boundary layer are calculated using XFOIL (Drela and Giles, 1989), a panel method that incorporates a modified version of the eN transition model
35    (van Ingen,1956). For the introduction of those transition points in the CFD simulations, a modified version of the 4 equations SST-Transition turbulence model is used. The intermittency is switched from 0 to 1 at XFOIL's calculated transition points locations to force the transition from a laminar to a turbulent boundary layer.

7. **Line 113: When you use a wall model for the rough surface part, what wall boundary condition do you use for the rest part of surface? Do you use a similar mesh resolution for both parts?**

40    The *nutkWallFunction* wall function is set for the clean airfoil part, whereas for the rough part the modified *nutkRoughWallFunction* wall function based on Cebeci (1977) is set (*nutkRoughWallFunction* condition inherits the traits of the *nutkWallFunction* boundary condition. ). The parameters $k_s$ and $C_s$ are defined to set the roughness wall functions. $k_s$ is the sand-grain roughness height (0 for smooth walls) and $C_s$ is the roughness constant with values from 0.5 to 1.

These modified wall functions are applied when the y+ has a value greater than 11.25. To avoid the uncertainty of being at the buffer layer when the model applies logarithmic wall functions, the meshes used in this work are designed with a larger cell height along all the airfoil at the wall in order to obtain a y+ of around 30. All this information has been added to the manuscript to make it clearer.

8. **The stall part of the lift and drag forces is not captured. Any suggestion for improvement?**

Thanks for the comment. In the conclusions of the paper, the last one refers to the uncertainty regarding the CFD simulations for high angles of attack. In this work the simulations were performed with steady state solver (simpleFoam) but the flow for high angles of attack is unsteady. An improvement will be obtained if a transitory solver is used in the simulations (pimpleFoam), however the authors decided to use the faster steady state simulations that reproduce correctly the airfoil performance at angles of attack used in normal production case used to estimate the annual energy production in clean/rough/eroded blades with and without VGS. For the sake of clarity the last conclusion of the paper that refers to this point has been expanded.

9. **In the rough cases, what turbulence model did you use?**

The SST k-w turbulence model was selected for the rough cases computations. In order to make it clearer, we have modified the manuscript on the following way:

> On the other hand, for the rough cases performed with the SST k-w turbulence model, the y+ values close to the airfoil surface are around 30, these y+ values are needed for the correct performance of the modified rough wall functions.

10. **In the erosion cases, you need to make a mesh sensitivity study as the small erosion parts are not easy to be captured as there are no measurement data to compare with.**

This is an interesting point to be addressed in all CFD studies, specially when simulating eroded blades, thanks for the comment. During the performance of this work, many effort has been done to mesh accurately the erosion types maintaining a balance between accuracy and efficiency in the simulations (not too loaded meshes). That is the reason why ANSYS was used for meshing the eroded blades instead ICEMCFD that was used for he 2D baseline airfoils. A sensitivity study was performed giving special attention to the size of the fist cell close to the wall to achieve the y+ values needed for the accurate simulation of the effect of erosion on blade sections performance. This study was not included in the first version of the manuscript since the main goal is to demonstrate the AEP recovery due to the use of VGs. A summary of the sensitivity analysis will be included in the second version of the manuscript.

11. **Many captions are miss-leading. For example, in Figure 8, in the first two cases it is indicated with the turbulence model, but not in the other cases.**

We fully agree with the reviewer, all graphs' captions have been modified and the turbulence model is indicated for all the simulated cases. Thanks for the comment.

12. **Section 4 on AEP is extremely short. Some details about the code and blades are required. What wind conditions do you consider? How do you use the 2D CFD results for rotor computations? Any corrections were performed?**

75      Thanks for the suggestion, we appreciate it and the section has been expanded in the manuscript adding a more detailed description of the code BladeOASIS and including a reference to a publication in which the code has been described. In addition the wind condition will explained: stationary wind is used and the Weibull parameters are c=7.5 m/s and k=2. In addition, the process used to include the CFD results has been elaborated in detail (mainly the computed polar curves are used in the BEM part of BladeOASIS). The Glauert tip loss correction used has been used.

---

## Author Comment (AC3)

**Reply on RC2**

David Bretos-Arguiñena[1] and Beatriz Méndez-López[1]

[1]Wind Energy Department, Centro Nacional de Energías Renovables (CENER), Sarriguren, Spain

**Correspondence:** David Bretos-Arguiñena(dbretos@cener.com)

We thank the reviewer for the time invested reviewing the manuscript and for the comments made. They will enhance the quality of the manuscript. The paper has been changed attending the referee suggestions, the typo errors corrected and more descriptions, tables and figures added. The answers to the different questions are explained in detail in the following pages.

1. **The manuscript contains many unclear and linguistically incorrect formulations and needs to be completely revised.**

   Thanks for your suggestion, the manuscript has been thoroughly revised in accordance with all the reviewer's comments.

2. **Abstract: The abstract is essentially a short summary and only the last sentence mentions a finding from the study. The presentation of the findings should be strengthened. In the abstract the calculation method (steady RANS?) should also be mentioned and it should be explained which "different technologies" are used in the calculations. Furthermore, it should be clarified in which respect the study goes "one step further".**

   We agree with the reviewer and the comments are welcome. The abstract has been amended as follows:

   > Blade roughness depositions or blade erosion have a negative impact on blade aerodynamics and wind turbine

power production. In this work, the effect of roughness, erosion and different flow conditions on the Annual Energy Production (AEP) of a standard wind turbine using the $NACA$ $63_3418$ airfoil at blade's tip has been studied. Computational Fluid Dynamics (CFD) steady-state computations with the OpenFOAM (v8) open source code have been performed. Different RANS turbulence models have been considered to emulate the studied conditions. Two types of erosion were considered, pits (type 1) and an extreme material loss (type 2); these were calculated as 3D simulations using hybrid meshes in order to reduce the high computational cost of using large meshes. Comparisons between structured and unstructured meshes to validate the use of hybrid meshes were also performed.

After studying the previous conditions, a pair of CENER's designed low drag vortex generators (VGs) were located in the upper side of the airfoil and were compared with conventional shape vortex generators. CENER's non conventional VGs were shown to decrease the airfoil's added drag and increase it's efficiency more than the conventional VGs studied. Having seen the advantages of using low drag vortex generators, it's effect on turbulent flow and, on a rough and eroded blades was studied. CENER's low drag VGs were shown to mitigate the harmful effect of roughness and erosion.

The obtained polar curves were introduced into CENER's in-house BEM tool BladeOasis, to translate the effect of these low drag VGs on blade aerodynamics in terms of impact on a 2.5 MW standard wind turbine AEP. The novelty of the work presented in this paper is that AEP losses due to the existence of blade surface roughness or erosion can be recovered from the use of VGs up to 1.5%. In this work, a methodology has been developed to introduce VGs in clean or eroded surface airfoils simulating accurately the boundary layer both in the VG surface and the airfoil surface without creating computationally expensive meshes.

3. **Line 7: Conventional VGs should be further specified (height, shape, . . . ). In the remainder of the manuscript, conventional VGs are not distinguished regarding their shape although they have specific characteristics according to the current state of research.**

We completely agree with the reviewer. The specifications of CENER's low drag VGs and the Conventional VGs are now indicated at the *"CENER's low drag VGs performance comparison vs conventional VGs"* subsection as follows:

**Table 1.** Considered VGs specifications.

| Characteristics | CENER's low drag VGs | Cropped VGs | Delta VGs | Rectangular VGs |
|---|---|---|---|---|
| VG height | 5 mm | 5 mm | 5 mm | 5 mm |
| VG chord | 12.5 mm | 12.5 mm | 12.5 mm | 12.5 mm |
| AoA | 18 ° | 18 ° | 18 ° | 18 ° |
| VG pair TE separation | 12.5 mm | 12.5 mm | 12.5 mm | 12.5 mm |

4. **Line 16: Some references on VG studies done in the last 20 years should be provided.**

Thanks for the suggestion, new references has been added to the introduction section.

5. **Line 21: In which way does roughness modify the flow characteristics of a fully turbulent flow? I would rather use "impacts" or "affects" instead of "modifies".**

This line leads to misunderstanding, it has been omitted. The manuscript in that part would be as follows:

> Roughness distribution on airfoil surface affects its performance by degrading the airfoil aerodynamic behaviour. If rough elements are located near the leading edge zone, laminar to turbulent transition upper and lower points are displaced towards the leading edge.

6. **Line 22f: Can the authors provide evidence (in form of references) that roughness elements act like first stages of blade erosion?**

Thanks for the suggestion, we agree that this sentence could be a bit confusing. To improve the quality of the work, the manuscript has been changed and the sentence modified to:

> The deposition of big roughness elements on the blade surface is considered in some studies as blade erosion. Ehrmaan et al (2017) described one possible aerodynamic explanation for wind turbine under performance is blade roughness caused by erosion (sand, salt, and hail), foreign deposits (insects, ice), or coating spallation. This is the reason why in this study roughness is included with erosion as one of the blade situations to be further studied, in addition since computational model set up is different in the roughness modelling and in erosion modelling, its inclusion widens the understanding of the effect of these phenomena

7. **Line 29: In the manuscript often just the name of the CFD code is mentioned (L.35, L.47). The authors should also mention the method used (steady-state RANS? turbu-lence model?) to enable the reader to better assess the work. The term "CFD" is too general.**

We completely agree with the reviewer, due to that reason a new Section has been added before the results. It explains in more detail the CFD settings. The new Section is:

The OpenFOAM CFD code was employed for the presented simulations. Steady-state time scheme and RANS (Reynolds-Averaged Navier Stokes simulations) turbulence models were used for all the computations performed. The incompressible solver *simpleFoam* based on SIMPLE algorithm (Semi-Implicit Method for Pressure-Linked Equations) of Caretto et al. (1973) was used.

The selected solvers and smoothers for solving the different terms equations are shown in the Table 2.

**Table 2.** Selected solvers and smoothers.

| Equation | Solver | Smoother |
|----------|--------|----------|
| $p$ | Geometric-algebraic multi-grid (GAMG) | Diagonal incomplete-Cholesky (DIC) |
| $U$ | | |
| $k$ | | |
| $\omega$ | Solver using a smoother (smoothSolver) | Gauss-Seidel (GaussSeidel) |
| $kl$ | | |
| $kt$ | | |

In this article different turbulence models have been used in order to consider different flow conditions, these turbulence models are introduced in the following sections. However, the assigned boundary conditions associated with the different variables employed by each model are shown in Table 3 and their meaning is explained at Appendix **??**.

**Table 3.** Assigned boundary conditions for the different performed simulations.

| Patch | $U$ | $p$ | $k$ | $\omega$ | $nut$ | $kl$ | $kt$ | $im$ |
|-------|-----|-----|-----|----------|-------|------|------|------|
| Farfield | freestream | freestream | inletOutlet | inletOutlet | calculated | fixedValue | fixedValue | inletOutlet |
| smoothAirfoil | fixedValue | zeroGradient | wallFunction | wallFunction | wallFunction | fixedValue | fixedValue | fixedValue |
| roughAirfoil | fixedValue | zeroGradient | wallFunction | wallFunction | roughWallFunction | fixedValue | fixedValue | fixedValue |

In the different simulations a fixed number of iterations was established, in order to achieve convergence for high angles of attack, since these are the most problematic ones. For the convergence criteria, the stabilisation of aerodynamic coefficients ($C_l$, $C_d$ and $C_m$) was used. The main reason to choose this criterion is to avoid the early ending of the simulations while the aerodynamic coefficients keep still changing due to a bad selection of residuals' values, and also to avoid misleading cases of false convergence due to excessively low relaxation factors. Once the different coefficients values keep oscillating around the same value, the simulations were considered converged.

Although residuals were not considered as convergence criteria, they were reviewed at some simulations (Fig. 1) in order to do not omit any strange behaviour. The obtained residuals are normalized ($L_1$ norm) and scaled (the initial residual values is 1). Residuals drop varies depending on the angle of attack, the larger the angle of attack is, the lower the residuals drop is. Regardless of the size of residuals' drop, the aerodynamic coefficients converge. For high angles of attack the aerodynamic coefficients oscillate remarkably due to the non stationary flow nature.

[Figure]

**Figure 1.** Residuals and aerodynamic coefficients evolution during the simulations for angles of attack of 0º, 12º and 20º; and the SST k-w turbulence model.

8. **Line 32: The statement of a power reduction of 6-7% is too general. For which roughness height was this value determined and was a fully turbulent flow or a natural transition scenario considered?**

   We thank the reviewer the suggestion, that study was made using a natural transition scenario, the line in the manuscript has been modified as follows:

   > Standish et al. (2012) performed experiments and simulations of a non specified airfoil of 18% thickness. The

55

> simulations were made using ANSYS CFX and roughness effects in a natural transition scenario were accounted for by introducing an equivalent sand-grain roughness on surface patches. Three different roughness cases were computed, 0.494 mm was the highest roughness considered. Afterwards AEP was calculated to evaluate the effect of roughness on power production. Normalized mechanical power output relative to measured power at two operational points below rated power of the SWT-2.3-93 wind turbine lead to power output reduction up to 10%, considering this an extreme case highly unlikely to happen.

9. **Line 37: on the field. Does this refer to field measurements or investigations in the area of simulating roughness effects**

We agree with the reviewer that the sentence is not clear enough, it has been modified as follows:

> More recently, further research in the area of CFD roughness effect simulations has been done.

10. **Line 45: Please insert a reference for the Knopp model.**

Thanks for the comment, the reference has been included in the manuscript.

11. **Line 51: What was varied with the 1000 different protuberances? Height, position, spacing, shape? The sentence is grammatically not correct.**

We thank the reviewer the suggestions, the text has been modified as follows:

> Kruse et al. (2021) evaluated the NACA $63_3418$ airfoil with 1000 different protuberances in the leading edge of the airfoil, varying the position and angle of edges, and the disturbances' height/depth. The main conclusion was that position and the depth/height of the disturbance were the most important parameters, with up to 35% lift reduction and 90% lift/drag reduction within the specified angle of attack (AoA) and disturbance parameter ranges.

12. **Line 64: The reference is not up to date, compare current status on WES (published 2023).**

Thanks a lot for the review, the year has been updated in the manuscript.

13. **Line 64/65: What are combined transition and rough CFD simulations? What does actual eroded blade mean? Was the actual shape of the leading edge erosion measured and taken into account in the CFD simulation? Which transition model was calibrated?**

We agree with the reviewer that the actual explanation of Vimalakanthan et al. (2023) work isn't clear enough, in order to make it clear and answer the reviewer's questions, the sentence have been modified as follows:

> Vimalakanthan et al. (2023) calibrated a modified version of the Langtry-Menter $SST\ k-\omega$ turbulence model

based on Langel et al. (2017b) considering laminar to turbulent transition and roughness effect on the order of 0.1-0.2 mm. Calculations of NACA $63_3418$ airfoil for clean flow and rough leading edge were presented. Moreover, CFD simulations of high-resolution scanned leading edge surfaces from an actual damaged blade were carried out. The conclusion of the study was that the calibrated CFD model for modeling flow transition accounting roughness shows good agreement of the aerodynamic forces for airfoils with leading-edge roughness heights in the order of 140-200 $\mu$m when comparing with the experiments, while showing poor agreement for smaller roughness heights in the order of 100 $\mu$m. The study indicated that up to 3.3 % reduction in AEP can be expected when the LE shape is degraded by 0.8 % of the chord, based on the NREL 5 MW wind turbine.

14. **Linie 70ff: The wording implies that VGs were developed for erosion problems. This sentence should be revised.**

Thanks for the comment, the sentence has been rewritten to improve its understanding and a reference to a recent review to flow control devices added.

In addition, flow-Control Devices for Wind-Turbine performance enhancement. In their review Akhter and Omar (2021) summarizes that flow-control mechanisms feature the ability to effectively enhance/suppress turbulence, advance/delay flow transition, and prevent/promote separation, leading to enhancement in aerodynamic and aeroacoustics performance, load alleviation and fluctuation suppression, and eventually wind turbine power augmentation. One of these types of control devices are VGs. VGs could be also used as mitigation measure for blade roughness or erosion.

15. **Line 73/74: Please add a reference for this statement.**

Thanks for the comment, the sentence has been changed in the manuscript and a new reference has been added to clarify its meaning.

Madsen, J.: Advances in aerodynamics of wind turbines blades, Wind Energy Denmark, Annual Event 2017

16. **Line 74: I suggest: streamwise vorticity**

Thanks for the suggestion 'stream vorticity has been used'.

17. **Line 74: on the blade surface → within the blade boundary layer**

Thanks, change done.

18. **Line 77/78: Increment in drag compared to what? Compared to a natural transition scenario?**

Increment in drag comparing an airfoil with leading edge roughness with and without VGs compared with the transitional case scenario. The mentioned line has been modified to make it clearer:

It was concluded that the increase in drag caused by VGs was negligible compared to that already caused by roughness with regard to the natural transition scenario.

19. **Line 78/79: Sentence is unclear. Is meant that drag is not increased when VGs are placed on the lower side (which would surprise me) or is meant that the drag of the lower side is not increased by VGs attached to the upper side? The dot at the end of the sentence is missing.**

Thanks for the comment, the sentence has been changed to make it more understandable. The authors of that work concluded that drag is not increased when VGs are placed on the lower side, it is literally mentioned in that paper:

" VGs on the pressure side ended up to be a suitable solution as for some angles of attack there is a lower CD value compared to rough cases due to the stalled flow suppression on the pressure side. This improvement would not be possible if that need was not discovered and VGs would be used only on the suction side "

20. **Line 79-81: The authors should describe in some more detail how Hansen et al. designed their aerodynamically shaped VGs and why the efficiency increase is spread over such a wide range. As a part of the literature review of the present manuscript it should additionally be mentioned that the (L/D)max of the clean airfoil is significantly higher than that of the airfoil equipped with VGs (even for the aerodynamically shaped ones).**

Thanks for the suggestion, the sentence has been modified and improved in the manuscript in this way:

> Moreover, aerodynamically shaped VGs have been designed and rated with regard conventional VGs trying to reduce the drag penalty added by any types of VGs to the blade surface compared with clean blades. For instance, Hansen et al (2016) proposed an aerodynamically shaped VG designed from a CLARK-Y airfoil with a thickness to chord ratio of 11.7%. It was added to a DU-91-W2-250 airfoil and tested in the wind tunnel. And an efficiency increase was observed with regard to conventional vortex generators from 3.6 % (for inflow angle 12°) to 16.36 % (for inflow angle 9°). This is due to the fact that in conventional VGs separation occurs right at the VG leading edge. The separation is expected to be smaller for the VGs based on airfoils and therefore the drag is believed also to be smaller.

21. **Line 82: Dot missing at end of sentence. Please use Bay or BAY consistently. What are the advantages of the fully resolved simulation of VGs and why did the authors choose this approach?**

Thanks for the suggestion, the missing dot has been added, BAY has been used instead of Bay and the answer to why the authors selected the fully resolved approach is included (in the section where the VGs simulations are described).

> With regard to simulation methodologies, when simulating VGs both CFD fully resolved and modelling approaches

are used. The fully resolved approach could be very expensive computationally since both the boundary layer at the airfoil and the VG walls should be accurately modelled which is only possible with expensive meshes. An alternative way of modelling VGs in CFD is to model the influence of the vortex generator on the boundary layer using body forces. In that sense, Bender et al (1999) presented a model for simulating the vane vortex generators without the necessity to define the VG geometry in the mesh. This model avoids the need of generating large and complex grids around the vane geometry by introducing a source term in the discretized momentum and energy equations. Another vortex generator model called jBAY was introduced by Jirasek (2005) for simulations of flow systems with VG arrays. The jBAY model is based on the lifting force theory of Bender E.E., Anderson B.H. and Yagle P.J. (1999) but with an improved technique for defining the model control points. Seel et al (2022) presented a comparison of performance of the use of BAY model versus the fully resolved approach when computing VGs. The agreement in terms of lift and pressure distribution is very good whereas the drag is underestimated by the BAY model.

22. **Line 82: The co-author published a study on low-drag VGs in 2018. Why was this paper not referenced and used as a basis for the present study? (doi: 10.1088/1742-6596/1037/2/022029)**

Since the VGs part has been expanded in the Introduction section we have added the reference to that paper as well as the main conclusions. Thanks for pointing this out.

23. **Line 86: Please be more precise. The power estimation was performed with BEM using (2D? calculated or measured?) polars for the blade sections without and with VGs/gurneys.**

We thank the reviewer for the suggestion and it is implemented in the manuscript as follows:

Regarding the evaluation of the effect on the production of introducing VGs into the blades, Fernandez-Gamiz et al. (2017) studied in their work the energy production increase of NREL 5 MW wind turbine using the blade element momentum theory and experimental polar curves from Timmer and Rooij (2003) and concluding that an overall increase of 3.85 % on the average wind turbine power output can be found when using VGs and gurney flaps are used. These VGs where located in the blade root and mid span area and no erosion or surface roughness were included in the simulations.

24. **Line 89//90: Please revise this sentence.**

The sentence has been reviewed. Thanks for the suggestion.

Skrzypinski et al. (2020) studied the effect on production of using VGs on wind turbine blades affected by surface roughness and predicted a gain of 3.3% experimentally and 2.8% using a predictive engineering tool, Power Pack, that is based in two engineering models for predicting the effect of roughness and VGs, three-dimensional effects and blade element momentum theory.

25. **Line 91ff: Please refer to the section numbers of your manuscript.**

Thanks for the suggestion, the manuscript has been modified as follows:

> This study is organized as follows: first, the CFD settings used are presented in Section 2. Next, CFD simulations of the NACA $63_3418$ airfoil in several surface conditions (clean, rough, pits and extreme loss of material) are shown in Section 3. Then, CENER´s low drag VGs are described and the CFD simulations of blade sections for the studied surface conditions including VGs are presented in Section 4. Finally, the annual energy production of a reference 2.5 MW wind turbine is evaluated in Section 5 to define the effect of roughness or erosion over AEP and to estimate how much of these losses can be recovered by the installation of low drag VGs over the blade surface.

26. **Line 100ff: The description of the numerical model is considerably too scarce to evaluate its suitability for the present analyses. How many grid cells were introduced across the boundary layer height? Number of total grid cells, mesh topology, far field distance, turbulence model, steady RANS or unsteady URANS, numerical scheme...? In the remainder of the manuscript unstructured and structured meshes are mentioned and a comparison is shown in Fig. 2. To the reader it is not clear which calculations documented in the manuscript were performed with which mesh type.**

Thanks for the detailed evaluation. This information has been added to the manuscript in Section 2 where the CFD simulations setup has been described in detail.

27. **Line 101ff: in clean $\rightarrow$ for clean; in the airfoil $\rightarrow$ along the airfoil; throughout the manuscript: y+ $\rightarrow y^+$; turbulent model $\rightarrow$ turbulence model; ICEMCFD $\rightarrow$ ICEM CFD**

Thanks for the revision, everything has been changed in the manuscript.

28. **Line 103: Please first mention that a wall model is used for the rough case, then the requirements for $y^+$.**

Thanks for the suggestion. The definition of the rough wall function has described more in detail in the manuscript.

> The *nutkWallFunction* wall function is set for the clean airfoil part, whereas for the rough part the modified *nutkRoughWallFunction* wall function based on Cebeci (1977) is set (*nutkRoughWallFunction* condition inherits the traits of the *nutkWallFunction* boundary condition. ). The parameters $k_s$ and $C_s$ are defined to set the roughness wall functions. $k_s$ is the sand-grain roughness height (0 for smooth walls) and $C_s$ is the roughness constant with values from 0.5 to 1.
>
> These modified wall functions are applied when the y+ has a value greater than 11.25. To avoid the uncertainty of being at the buffer layer when the model applies logarithmic wall functions, the meshes used in this work are designed with a larger cell height along all the airfoil at the wall in order to obtain a y+ of around 30.

29. **Line 108 and throughout the manuscript: eN $\rightarrow e^N$, k-kl-w $\rightarrow k - kl - \omega$. Remark: In XFOIL a simplified $e^N$ envelope method is implemente d, in which the frequency dependence of the TS amplification is not considered, in contrast to the original method proposed by van Ingen.**

Thanks for the suggestion and the remark. Everything has been changed in the manuscript.

30. **Line 113:** $5e^{-3} \rightarrow 5 \cdot 10^{-3}$**. This should be used consistently throughout the manuscript.**

Thanks for the revision, it has been changed in the manuscript.

31. **Line 118: The experiments are well reproduced by the calculation except for the stall domain.**

Thanks for the suggestion. That observation is made in manuscript and in the conclusions section.

140  32. **Caption Fig. 1 and Fig. 2: The CFD results are only shown as lines, so that the chosen angle-of-attack discretization is not clear. I suggest to add some symbols.**

Thanks for the suggestion. The manuscript's figures have been modified using lines and points to make clearer the AoA discretization.

33. **Caption Fig. 2: eN turbulence models. The $e^N$ method is not a turbulence model but rather a transition prediction**
145  **model.**

Thanks for the suggestion, the caption has been modified as follows:

[Figure]

**Figure 2.** Lift, drag, efficiency and polar curves for the clean cases with $k - kl - \omega$ turbulence model and $e^N$ transition model. Experiments for clean case included. NACA $63_3418$ $\mathrm{Re} = 3 \cdot 10^6$.

.

34. **Fig. 3: In the legend (XFOIL) I suggest to write transition location upper / lower side (XFOIL).**

    Thanks for the suggestion, the manuscript has been changed accordingly.

35. **Line 130: shows the pressure distribution or shows the distribution of the pressure coefficient**

150      Thanks for the suggestion it shows the distribution of the pressure coefficient, the manuscript has been changed accordingly.

36. **Line 149: SST k-w $\rightarrow SST\ k-\omega$. Was this turbulence model also used in the previous calculations with prescribed XFOIL predicted transition location? Then this should be mentioned in Sec. 2.1.**

    Thanks for the suggestion, yes, it is the $SST\ k - \omega$ and it has been mentioned in Sec. 2.1.

37. **Figs. 10/15/16/18: Are the velocities in the first grid layer shown in the top views? Is the axial velocities component shown or the absolute value of the velocity? The legends in the top views are too small and partly blurry.**

Thanks for the comment. The top view show the streamlines but not in the first grid layer. The value shown is the absolute value of the velocity. The legends are changed and made bigger. All this has been added to the manuscript.

38. **Table 1 / Fig. 8: ($\alpha$ in [°], $\Delta(Cl/Cd)_{max}$?!). Obviously, the calculations were only performed up to an angle of attack of 14° and at least for the clean and the turbulent case without erosion, the theoretical cl, max values are obviously not yet reached. This angle should therefore not be listed as $\alpha$(cl, max) in Table 1.**

Thanks for the comment, to make the results more comparable, Tables 1 and 2 have been changed and the maximum Cl column is substituted by the Cl at 10 degrees. They are added to the manuscript and to this review.

**Table 4.** Summary of airfoil aerodynamic characteristics. CFD simulations and experiments NACA $63_3418$ $Re = 3 \cdot 10^6$.

|  | $\mathbf{C_{l,10°}}$ | $\mathbf{C_{d,min}}$ | $\alpha_{\mathbf{C_{d,min}}}$ | $\mathbf{(C_l/C_d)}_{max}$ | $\alpha_{\mathbf{(C_l/C_d)_{max}}}$ | $\mathbf{\Delta(C_l/C_d)}$ [%] |
|---|---|---|---|---|---|---|
| CFD Clean (k-kl-w) | 1.49 | 0.58e-2 | -1 | 116.96 | 5 | - |
| CFD turbulent flow (SST k-w) | 1.35 | 1.01e-2 | 1.60 | 54.54 | 6 | -53.37 |
| CFD P40 (423$\mu$m) 15 %c | 1.06 | 1.43e-2 | -1 | 36.80 | 5 | -68.54 |
| CFD Erosion Typology 1 | 1.28 | 1.33e-2 | 0 | 41.98 | 6 | -64.11 |
| CFD Erosion Typology 2 | 1.02 | 1.59e-2 | 0 | 31.30 | 4 | -73.24 |
| Experimental Clean | 1.23 | 0.64e-2 | 0 | 118.00 | 5 | - |
| Experimental P40 (423$\mu$m) 15 %c | 0.98 | 1.48e-2 | 0.98 | 38.14 | 5 | -67.6 |

**Table 5.** Summary of section aerodynamic characteristics. CFD simulations without and with VGs. NACA $63_3418$ $Re = 3 \cdot 10^6$.

|  | $\mathbf{C_{l,10°}}$ | $\mathbf{C_{d,min}}$ | $\alpha_{\mathbf{C_{d,min}}}$ | $\mathbf{(C_l/C_d)}_{max}$ | $\alpha_{\mathbf{(C_l/C_d)_{max}}}$ | $\mathbf{\Delta(C_l/C_d)}$ [%] |
|---|---|---|---|---|---|---|
| 1a CFD turbulent flow | 1.35 | 1.01e-2 | -2 | 54.54 | 6 | - |
| 1b CFD turbulent flow + VG | 1.51 | 1.05e-2 | 0 | 56.02 | 6 | +2.71 |
| 2a CFD P40 15%c | 1.06 | 1.43e-2 | -1 | 36.80 | 5 | -32.53 |
| 2b CFD P40 15%c + VG | 1.46 | 1.30e-2 | 0 | 44.40 | 6 | -18.59 |
| 3a CFD Erosion Typology 1 | 1.28 | 1.33e-2 | 0 | 41.98 | 6 | -23.03 |
| 3b CFD Erosion Typology 1 + VG | 1.46 | 1.22e-2 | 0 | 47.15 | 6 | -13.55 |
| 4a CFD Erosion Typology 2 | 1.02 | 1.59e-2 | 0 | 31.30 | 4 | -42.61 |
| 4b CFD Erosion Typology 2 + VG | 1.23 | 1.40e-2 | 0 | 34.43 | 4 | -36.87 |

39. **Line 167f: VGs only increase cl in case of flow separation for the baseline airfoil, and particularly increase cl, max and $\alpha$(cl, max). They do not increase lift for the complete AoA range. This should be expressed more clearly.**

We agree with the reviewer and in order to make it clearer, the suggested paragraph is modified as follows:

> VGs are passive flow control devices that are usually triangular or rectangular vanes inclined to the flow and are sized with regard to the local boundary layer thickness. VGs are known for their capability to delay separation and increase lift force on the blade. It's effect is notable for high angles of attack where flow separation occurs, and the maximum lift coefficient is increased.

40. **Line 168: I suggest: . . . added to the blade for attached flow.**

Thanks for the suggestion, the manuscript has been changed accordingly.

41. **Line 169ff: CENER proposes VGs whose cross-section feature a 10% thick aerodynamically shaped airfoil. Given the very small Reynolds numbers of the VG, flat plates are actually not a bad choice and provide well defined separation locations for the generation of the streamwise vortices. The authors initially claim without evidence that their aerodynamically shaped VG cross section show lower drag than conventional VGs, but it is not clear whether this statement refers to the self drag of the VGs or to the drag of the blade section equipped with these VGs. The authors do not provide an explanation or a reference. Fig. 14 then seems to support the claim. It is surprising, however, that the airfoil with the proposed VGs shows a significantly lower drag than the turbulent airfoil without VGs, even at small angles-of-attack. This is not plausible to me and is also in contrast to previous studies. For example, it was shown by Hansen et al. (doi: 10.1002/we.1842) that for the (aerodynamically shaped) VGs they studied, the L/Dmax of the blade section with VGs is reduced. In the present manuscript a sound flow-physics analysis and a physical explanation of the beneficial effect of the proposed VGs is urgently needed! Does the turbulent airfoil without VGs show flow separation even at small angles of attack, which are suppressed or delayed by the VGs? If so, how can it be explained that the conventional VGs do not suppress the separation? Are mean momentum and displacement thicknesses of the boundary layer reduced by the VGs? Here, an analysis of the boundary layer development without / with VGs as well as an analysis of the flow field and of skin friction and pressure drag components is necessary.**

Thanks for the big comment. The answer is addressed step by step.

COMMENT 1: The authors initially claim without evidence that their aerodynamically shaped VG cross section show lower drag than conventional VGs, but it is not clear whether this statement refers to the self drag of the VGs or to the drag of the blade section equipped with these VGs.

ANSWER 1: The sentence refers to the drag penalty added to the blades sections equipped with these VGs. The sentence in the manuscript is changed accordingly. In addition, in previous works Mendez (2018) and Hansen (2016) showed that aerodynamically shaped VGs have a lower self drag due to their aerodynamic shape.

COMMENT 2: It is surprising, however, that the airfoil with the proposed VGs shows a significantly lower drag than the turbulent airfoil without VGs, even at small angles-of-attack. This is not plausible to me and is also in contrast to previous studies

ANSWER 2: The coefficient values for the different angles of attack and airfoil configurations shown in Figure 14 has been compared in the following table. This table has been added to the manuscript. At it is observed Cd increases for all cases when adding VGs with the exception of the Turbulent case + Low Drag VGs for an angle of attack of 0°. This case is being analysed more in detail in order to find an explanation to the issue.

COMMENT 3: In the present manuscript a sound flow-physics analysis and a physical explanation of the beneficial effect of the proposed VGs is urgently needed! Does the turbulent airfoil without VGs show flow separation even at small angles of attack, which are suppressed or delayed by the VGs? If so, how can it be explained that the conventional VGs do not suppress the separation? Are mean momentum and displacement thicknesses of the boundary layer reduced by the VGs? Here, an analysis of the boundary layer development without / with VGs as well as an analysis of the flow field and of skin friction and pressure drag components is necessary.

ANSWER 3: Thanks for the comment. We fully agree that when a purely VGs investigation is made a detailed flow analysis should be made, evaluating the boundary layer characteristics with and without VGs as well as the vorticity values evolution downwash the VGs. The essence of this work is to demonstrate that the use of VGs when the blade surface is affected by erosion or with roughness can recover the AEP losses and that it can be achieved with a CFD procedure that is computationally efficient. That is also the reason why the parametric study to select the best VG location is not included and neither the VG location is changed for the different surface status options. Single location and simple study for demonstrate that VGs are a valid mitigation option. In any case we will try to attend the reviewer comment, and expand the boundary layer description if we find the time in the weeks available to modify the manuscript.

**Table 6.** Summary of the calculated aerodynamic coefficients for turbulent flow conditions considering an airfoil without VGs and with CENER's low drag VGs, Cropped VGs, Delta VGs and Rectangular VGs.

| | Turbulent | | Turbulent+CENER's VGs | | Turbulent+Cropped VGs | | Turbulent+Delta VGs | | Turbulent+Rectangular VGs | |
|---|---|---|---|---|---|---|---|---|---|---|
| AoA | $C_l$ | $C_d$ | $C_l$ [%] | $C_d$ [%] | $C_l$ [%] | $C_d$ [%] | $C_l$ [%] | $C_d$ [%] | $C_l$ [%] | $C_d$ [%] |
| 0 | 3.25E-01 | 1.07E-02 | +9.54 | -1.87 | +2.15 | +19.63 | +1.85 | +18.69 | +1.23 | +13.08 |
| 2 | 5.47E-01 | 1.22E-02 | +8.96 | +0.82 | +3.47 | +19.67 | +2.93 | +19.67 | +2.93 | +22.13 |
| 4 | 7.64E-01 | 1.46E-02 | +9.03 | +6.16 | +4.45 | +22.60 | +4.32 | +23.97 | +4.19 | +23.29 |
| 6 | 9.73E-01 | 1.78E-02 | +9.97 | +7.30 | +5.86 | +25.28 | +5.86 | +25.84 | +5.86 | +26.40 |
| 8 | 1.17 | 2.21E-02 | +29.06 | +9.05 | +6.84 | +26.70 | +6.84 | +27.60 | +7.69 | +27.15 |
| 10 | 1.35 | 2.74E-02 | +25.93 | +12.04 | +8.89 | +29.56 | +8.89 | +29.93 | +8.89 | +29.56 |
| 12 | 1.49 | 3.53E-02 | +23.49 | +11.05 | +12.08 | +26.35 | +12.75 | +27.48 | +12.75 | +28.33 |
| 14 | 1.6 | 4.71E-02 | +15.00 | +11.89 | +15.00 | +19.96 | +15.00 | +29.72 | +16.88 | +17.83 |

42. **Line 174: The aerodynamic behavior of VGs depends primarily on the geometrical parameters of the VG array (i.e., inclination angle, interspacing, intraspacing, VG height with respect to the local GS thickness, VG length on the airfoil surface and at the VG tip, positioning of the VG pairs with respect to each other). This should be mentioned in the paper. If necessary, the influence of these parameters should also be investigated. Hansen et al. showed that the inclination angle still produces a large increase in efficiency. Is this also the case with the CENER VG?**

Thanks for the suggestion. The geometric description of the VGs has been included in the paper. The influence of these parameters has not been included in the paper since that study is out of the scope of this work. CENER Low Drag VGs were designed under the scope of the H2020 project leaded by ORE Catapult 'Offshore demonstration blade'. During the design phase, all the parameters´ influence over the VGs behaviour was made and included in the project´s deliverable. That information was not published in a research paper since the priority there was to test CENER´s low drag VGs in the wind tunnel. In fact, they were tested in the WindGuard Wind Tunnel in CENER´s airfoil family. Since this airfoil family is not public the wind tunnel measurements were only partially published in Wind Europe conference 2019.

43. **Fig. 14: In this graph, the difference between conventional (Delta, Cropped and Rectangular) VGs is not visible. Can you zoom in to show the differences? Differences of conventional VGs should be predicted in qualitative agreement to results of previous studies to make the predictions for the proposed VGs more plausible.**

Thanks for the suggestion. A table with the coefficient values has been included in the manuscript to make it clearer. Other studies, as the one suggested by the reviewer from Baldacchino (and evaluated in detail in point 46-line 197ff) demonstrated that the effect of VG shape on a 30% airfoil efficiency was very small. It is necessary to point out that the VGs studies depend on the airfoil shape, thickness, Reynolds numbers, VG geometry, location, and we agree that drawing general conclusions should be made with caution.

44. **Line 184: Was the boundary layer of the VGs actually resolved in the present calculations? What criteria were used to ensure accurate meshing of the VGs? It is unclear how a meaningful refinement in the VG region was achieved without increasing the total number of grid cells.**

Thanks for the suggestion. A detailed description of the boundary layer resolution and meshing is included in the manuscript. This is one of the strengths of this work, the capability of solving the VGs and blade boundary layer without increasing the total number of grid cells.

Hybrid meshes combining tetrahedrons and prisms have been used in order to obtain a refined mesh at VGs or eroded locations areas without extending this number of cells increment to the rest of the domain. The flexibility of this type of unstructured meshes allows for a more affordable computational cost. In the present calculations the boundary layer was resolved, the same meshing criteria as in the previous simulations was used.

45. **Line 197f: The parametric study of the chordwise VG position for rough conditions should be shown in this work. Since the boundary layer profiles are significantly changed by roughness, it may not be feasible to use best practices for clean airfoils. A detailed study of the boundary layer parameters would add scientific value.**

The parametric study was made using a modified version of XFOIL presented by Tavernier (2017) an selecting the best location for a conventional VG pair in the NACA $63_3418$ for the turbulent case configuration. It has not been included in the manuscript since it does not add any value and may lead to confusion since the same location has been selected for all configurations. This study is not a purely VGs study since it is a wider approach to the rough-erosion-VGs simulations an effect over AEP.

46. **Line 197ff: Regarding their efficiency the different types of conventional VGs are not distinguished in this statement, which does not reflect the current state of research (see Schubauer et al. https://doi.org/10.1017/S0022112060000372 or more recently Baldacchino et al. 10.1002/we.2191).**

In Baldacchino study they show the effect of different VGs configurations in the 30%-thick DU97-W-300 airfoil, and its performance was evaluated in the Delft University Low Turbulence Wind Tunnel at a chord-based Reynolds number of 2×106. It was observed and effect over airfoil efficiency varying from 56 for the counter-rotating common upwash configuration to 72 approximately for the co-rotating configuration. This was regard the configuration. With regard to the shape, the two different shapes evaluated changing the cropped length lead to a very similar efficiency value of 65.

The scope of the work that is now under revision, is not focused in a pure vortex generator study as the one from Baldacchino, it is an evaluation of the effect of using VGs in an 18% thickness airfoil with different surface conditions (erosion roughness) and the effect on AEP recovery due to their use.

In the manuscript, line 197ff has been review to make it clearer, thanks a lot for the observation.

47. **Linie 199f: Vortex formation of VGs is a complex topic, which cannot be covered by seemingly arbitrarily placed streamlines. More meaningful properties (velocity profiles in the wake, boundary layer parameters or cross-flow kinetic energy (see dissertation Florentie https://doi.org/10.4233/uuid:704d764a-6803-4cad-991f-45dc4ea38f6d)) should be used here.**

The vortex formation in VGs is a really complex topic. As described in previous answers, this work is not a comparison of conventional vs aerodynamic VGs, this work is an evaluation for a certain blade surface status of the effect of using Low drag VGs as mitigation measure. That is the reason why that complete study of velocity profiles, boundary layer parameters, etc was done in Mendez 2018 comparing conventional and aerodynamic VGs. In that study the aerodynamic VGs were base in ClarkY and RonCZ airfoils, the CENER Low drags VGs were analysed in the same way but not included in that study since they were being tested in the wind tunnel at that moment.

48. **Fig. 15: What is the meaning of this figure, the text does not contain any interpretation? Why is the flow field around the new VGs shown in a different way in Fig. 16? Here, a representation as in Fig. 15 a) b) c) and a direct comparison with the flow fields of the conventional VGs in connection with a flow-physics analysis would make**

**sense. What is the origin of the geometric parameters of the three configurations? Is the thickness of the VGs to scale?**

Figure 15 is a qualitative comparison of the vortex structure in conventional VGs. This figure has been completed and discussed in the manuscript adding a view of the vortex structure in several positions downwards of the VG pair. The figure is shown below. CENER´s Low drag VGs are not included in this figure due to confidentiality reasons. If a top view of the cross section of the VG is shown, its shape could be easily digitalized. The geometric parameters of the configurations are described in the table added in the answer to question 3.

[Figure]

**Figure 3.** Vorticity contours at sections 1c, 3c and 5c (c = VG chord) after the VG pair. VGs placed in the NACA $63_3418$ airfoil $Re = 3 \cdot 10^6$ AoA 12º.

.

49. **Fig: 16: The figure does not add value as it is not suitable for assessing the low drag VG. Quantitative analysis should be provided here, e.g. as presented by the co-author at Torque 2018 (doi :10.1088/1742-6596/1037/2/022029).**

Thanks for the suggestion. The figure has been improved adding the vortex structure right after the VG pair.

[Figure]

**Figure 4.** Vorticity contours at sections 1c, 3c and 5c (c = VG chord) after the VG pair. VGs placed in the NACA $63_3418$ airfoil Re $= 3 \cdot 10^6$ AoA 12º.

.

50. **Line 208/209: The statement "...recovery of the airfoil efficiency..." is exaggerated. The improvement in performance is actually very small. Furthermore, I cannot understand the values given in Table 2 for $\Delta(Cl/Cd)_{max}$. According to the table, config. 2a should show a L/D reduction of more than 32% compared to turbulent flow, which I cannot see in Fig. 17. I ask the authors to check and correct the values listed in table 2.**

The statement 'recovery of the airfoil efficiency' has been changed in the manuscript.

With regard to Table 2, there was an error in the plot, thanks for point it out. A wrong data set has been selected. It has been changed in the manuscript.

51. **Fig. 18 / 19 and associated text: How were the separation locations (65% and 75%) determined, for which spanwise range are the cf values shown in Fig. 19 and could you show spanwise averaged distributions? What is the reason for the strongly positive or negative cf values at 30% chord? How can a single small VG pair have such a strong influence on separation location of a blade section with such a large span? More generally, I also wonder why such a large span was considered in the calculations, but only one VG pair modelled. Would it not have made more sense to use a segment with a smaller width and periodic boundary conditions?**

Thanks for the complete comment. We will answer the questions step by step.

QUESTION 1: Fig. 18 / 19 and associated text: How were the separation locations (65% and 75%) determined

ANSWER 1: They have been determined identifying the location where the friction coefficient is cero.

QUESTION 2: which spanwise range are the cf values shown in Fig. 19 and could you show spanwise averaged distributions?

ANSWER 2: This figure has been changed and the friction coefficient has been averaged with the spanwise dimension to 'clean' the figure and make it more easy to interpret.

[Figure]

**Figure 5.** Skin friction coefficient for erosion Typology 1, erosion Typology 1 + VGs and turbulent cases. NACA $63_3418$ Re $= 3 \cdot 10^6$ AoA 12º.

.

QUESTION 3: What is the reason for the strongly positive or negative cf values at 30% chord?

ANSWER 3: The 30% is the VGs location. With the new Figure 19 the peaks have disappeared which helps to conclude that they could came from a numerical instability.

QUESTION 4: How can a single small VG pair have such a strong influence on separation location of a blade section with such a large span? More generally, I also wonder why such a large span was considered in the calculations, but only one VG pair modelled. Would it not have made more sense to use a segment with a smaller width and periodic boundary conditions?

310  ANSWER 4: The explanation to that question could be due to the fact that the blade section is eroded so the flow is very disturbed by the existence of erosion. The same study could be made comparing with the clean case with and without vGs. It is being analysed at this moment.

The large span was considered in the calculations since the erosion pattern is fully 3D. The suggestion of using several VGs pairs is very good and will be performed . If possible, we will include it in further manuscript versions.

315  52. **Line 226: What are the corresponding polar curves ?**

Thanks for pointing this sentence out. It has been written differently to improve its quality. In addition, this section has been enlarged to explain in more detail how the AEP has been evaluated for the different blade status configurations.

53. **Sec. 5: Conclusions consisting only of a list are unusual. Some of the aspects mentioned were not discussed in the manuscript and may not be included in the conclusions, for example line 247-249. Other listed conclusions actually**

320  **represent an outlook: line 255-256, line 262-264. The conclusion line 261-262 was not discussed and verified in the manuscript.**

Thanks for the suggestion. The conclusions Section has been rewritten to make it clearer attending the reviewer comments.